# Factors associated with low back pain among construction workers in Nepal: A cross-sectional study

Bikram Adhikari[1]*, Anup Ghimire[1], Nilambar Jha[1], Rajendra Karkee[1], Archana Shrestha[2,3,4], Roshan Dhakal[1], Aarju Niraula[1], Sangita Majhi[1], Antesh Kumar Pandit[1], Niroj Bhandari[2,4]

1 School of Public Health and Community Medicine, B.P. Koirala Institute of Health Sciences, Dharan, Nepal,
2 Department of Public Health, Kathmandu University School of Medical Sciences, Dhulikhel, Nepal,
3 Department of Chronic Disease Epidemiology, Yale School of Public Health, New Haven, Connecticut, United States of America, 4 Institute for Implementation Science and Health, Kathmandu, Nepal

* bikram.adhikariadhitya@gmail.com

## Abstract

**Data Availability Statement:** The data has been deposited to Open Science Framework (OSF) public repository with URL: https://osf.io/xb8k5/.

### Background

Low back pain (LBP) is the commonest cause of disability throughout the world. This study aimed to determine the prevalence and factors associated with LBP among the construction workers in Nepal.

### Methods

A community-based cross-sectional study was conducted among the construction workers working in Banepa and Panauti municipalities of Kavre district, from September 2019 to February 2020. Data was collected purposively by face-to-face interview from 402 eligible participants from the both municipalities using semi-structured questionnaire. Mobile-based data collection was done using KoboCollect. Data were exported to and analysed using R-programming software (R-3.6.2). Univariate and multivariate logistic regressions were performed. All tests were two tailed and performed at 95% confidence interval (CI).

### Result

One-year prevalence of LBP among construction workers were 52.0% (95%CI: 47.0–57.0). The higher odds of LBP was reported among females [adjusted odds ratio (aOR) = 2.42; 95%CI: 1.12–5.23], those living below poverty-line (aOR = 2.35; 95%CI: 1.32–4.19), participants with more than five years of work experience (aOR = 1.66; 95%CI: 1.01–2.73) and those with intermediate sleep quality (aOR = 2.06; CI: 1.03–4.11). About 80.0% of construction workers with LBP never seek healthcare services due to: a) time constraints (90.9%), b) financial constraints (18.1%) and c) fear of losing wages on seeking healthcare services (40.9%). The majority of the participants (94.8% among those without LBP and 72.3% among those with LBP) did nothing to prevent or manage LBP.

**Funding:** The author(s) received no specific funding for this work.

**Competing interests:** The authors have declared that no competing interests exist.

## Conclusion

The prevalence of LBP in the past one year was high among construction workers where majority of workers never did anything to prevent or manage LBP. Therefore, the public health professionals should set up the health promotion, education, and interventions aimed at increasing awareness on preventive techniques and predisposing factors of LBP.

## Introduction

Nepal is an agrarian society where majority of the population are dependent on works requiring huge physical capabilities such as farming. Majority of the population of Nepal have informal employment dependent on traditional agricultural practices. Even within the formally employed population, agriculture and construction works are the major areas of employment [1]. According to the Nepal labour force survey (2017/18) and Economic survey (2018/19), about 13.8% of Nepalese population are engaged into construction sector. It is estimated to constitute around 10.3% of the Gross Domestic Product (GDP) in Nepal [1,2]. Socio-economically poor people from the rural areas are engaged into this sector [3].

Construction work is renowned as unhealthy because of the high mechanical nature and hard physical labor involved into it [4,5]. A construction manual worker is a general/blue-collar worker employed in the construction industry and works predominantly on construction sites. They are typically engaged in hands-on aspects of the industry other than the design or finance. This includes members of specialist trades such as builders, electricians, carpenters, bricklayer, manual labor, armature fixing workers, internal finish workers and plumbers [6].

Construction workers are at a higher risk of musculoskeletal conditions with low back pain, a common problem among construction workers as nearly 80% working postures were found harmful for the musculoskeletal system of the construction workers [7–13]. Low back pain was found in about half of the respondents in the various studies among construction workers in Saudi Arabia, Sweden and India [11–13]. Many construction manual workers may suffer from low back pain but do not report it as an injury [14].

Low back pain (LBP) is the most prevalent musculoskeletal condition and a leading cause of disability throughout the world [15]. It is one of the least prioritized non communicable diseases in Nepal [16]. LBP causes disability, severe pain and extended sick leave affecting about 80% of individuals during their lifetime [17,18]. LBP leads to high direct and indirect costs which have great medical, social and economic impacts for individuals, families, society, and government [19–22]. A large proportion of Nepalese is engaged in the construction sector which has a bad reputation due to the high mechanical nature and hard physical labor involved. The health of the construction workers is an important issue that needs to be addressed for the development of the country. Little is known about LBP and factors associated with LBP among construction workers of Nepal. In addition, there are very few studies to understand about measures taken by construction workers to prevent and manage LBP.

Here, we aimed to estimate the prevalence of LBP among construction workers, associated factors of LBP (socio-demographic, lifestyle, occupational and psychosocial factors); and measures taken by construction workers to prevent and manage LBP.

## Methods

### Study area and study participants

We conducted a cross sectional study between September, 2019 and February, 2020 in Kavre district of central Nepal. Two out of 13 municipalities in Kavre district, Panauti and Banepa,

were purposively selected. We visited every ward of each municipalities and located building construction sites with the assistance of local people. We approached the construction workers available in the construction sites and those who meet eligibility criteria were enrolled into the study. The inclusion criteria included a) building construction workers aged 18 years or above and b) construction workers having a work experience of one year or more. The exclusion criteria of the study were: a) construction workers who were not able to communicate clearly (hearing and communication impaired) and b) those who did not provided any consent.

## Sample size

The sample size was calculated using Cochran's formula assuming the prevalence (p) of LBP among construction workers to be 21.0% based on the study by Reddy *et al* [23], 4.0% (20.0% of p) absolute error and 95% confidence interval(CI). With 5.0% non-response rate, final sample size calculated to be 419.

## Data collection tools and techniques

Data were collected through face-to-face interviews and anthropometric measurements.

**Interview.** Data collection was done using a semi-structured questionnaire which included the following components:

- A questionnaire assessing socio-demographic characteristics of participants like age, gender, religion, ethnicity, marital status, and average monthly family income; lifestyle characteristics and occupational characteristics.

- Extended Nordic Musculoskeletal Questionnaire was included to assess LBP. The validity of Extended Nordic Musculoskeletal Questionnaire (NMQ-E) was done by three experts: an epidemiologist, orthopedic surgeon, and community medicine expert.

- Questions addressing psychosocial factors such as job insecurity, work-family balance, job satisfaction and exposure to the hostile work environment.

- Depression Anxiety and Stress Scale (DASS-21) was used to assess the depression, anxiety, and stress.

**Anthropometric measurement.** The height and weight were measured using the digital weighing machines and the portable measuring tape, respectively, from which body mass index (BMI) was calculated. The BMI was calculated using the formula: weight (kg) / height (cm) $^2$.

## Variables

**Low back pain.** Low back pain (LBP) is said to be present if the participant report pain and discomfort localized below the costal margin and above the inferior gluteal folds, with or without leg pain [21].

**Poverty.** Those participants whose family members living with less than $1.90 per day were considered to be living below the poverty line [24].

**Obesity.** Participants were classified into Underweight, Normal, Overweight and Obese based on Asia-Pacific guidelines for obesity classification [25].

**Current smoker.** Current smokers were defined as those who reported smoking any tobacco product within the last 30 days. Respondents who reported smoking at least 100

cigarettes in their lifetime and who, at the time of the survey, did not smoke were defined as past smokers [26].

**Current alcohol drinkers.**   Those who consumed alcohol within the last 30 days were considered current alcohol drinkers [26].

**Sleep duration.**   Sleep duration was assessed by asking two questions-"What time you usually go to sleep at night? And what time you usually wake up in the morning?" Later based on the response, sleep duration was coded into four categories: (a) 3–4 hours, (b) 5–6 hours, (c) 7–8 hours, and (d) 9 hours and more (24).

**Sleep quality.**   It was measured by two questions—1) "In the past one year, how do you rate your own sleep quality?" The response were a) Good b) Intermediate c) Poor and 2) Do you think you sleep enough? The response was either yes or no [27].

**Co-morbidities.**   The participants were asked whether they had any kind of existing diseases like diabetes, hypertension, stones or others. Those who had existing diseases were considered for the presence of co-morbid conditions.

**Type of construction work.**   Based on several pieces of literature, the construction workers were classified into manual labors, bricklayer, plumber, electrician, painter, carpenter, interior finish worker, scaffolders and armature fixing worker.

**Employment status.**   It is to determine whether the employment status is seasonal or permanent.

**Work experience.**   Duration of work experience was assessed by asking two questions—1) At what age have you started working in the construction industry? and 2) How long have you been doing construction work?

**Depression, anxiety and stress.**   Depression, anxiety and stress were assessed using Depression, Anxiety and Stress Scale (DASS-21) developed by Lovibond *et al* [28]. The internal consistency of the DASS-21 Nepali version was 0.77 for DASS-depression; 0.80 for DASS-anxiety; and 0.82 for DASS-stress, which indicates good Cronbach's alpha values [29].

**Job satisfaction.**   It was measured by the questions- "Please tell me whether you: strongly agree, agree, disagree, or strongly disagree with this statement: I am satisfied with my job". Responses of "strongly disagree" and "disagree" were defined as low job satisfaction [30].

**Work-family imbalance.**   It was measured by the following question: "Please tell me whether you: strongly agree, agree, neutral, disagree, or strongly disagree with this statement: It is easy for me to combine work with family responsibilities." Responses of "strongly disagree" and "disagree" were defined as high work-family imbalance [30].

**Exposure to the hostile work environment.**   It was measured by the question "During the past 12 months were you threatened, bullied, or harassed by anyone while you were on the job?" The response of "Yes" was defined as exposure to a hostile work environment [30].

**Job insecurity.**   Job insecurity was measured by the question: "Please tell me whether you: strongly agree, agree, disagree, or strongly disagree with this statement: I am worried about becoming unemployed." Responses of "strongly agree" and "agree" were defined as high job insecurity [30].

## Data handling

The questionnaire was created in the platform of Kobotoolbox. Principal Investigator collected data with an android mobile phone using KoboCollect software and synchronized it into the cloud of Kobotoolbox.

Data collected into the mobile phone was monitored for its completeness after individual data entry. Proper validation and checks of the questionnaire was setup in the Kobotoolbox to provide assurance of uniform and correct data entry. Ten percent of the collected data were cross-checked for its cleanliness, errors, and problems at the end of every week.

The data synchronized into Kobotoolbox platform was exported into R-programming software (R-3.6.2) for cleaning, coding, categorizing and to check completeness, consistence and outliers.

## Statistical analysis

Data was analysed using R-programming software (R-3.6.2) after completing pre-analysis tasks. Descriptive analysis of socio-demographic and occupational characteristics and factors related to LBP was conducted. Means and standard deviations were calculated for normally distributed numerical variables and, medians and inter quartile ranges (IQR) otherwise. Frequency and percentages were displayed for categorical variables. The confidence interval around the prevalence was determined using the Clopper-Pearson method. Student's t test or Mann Whitney U test was applied to compare numerical data between two groups. Univariate and multivariate logistic regression analysis were carried out to determine the association of factors with LBP. Throughout the study, all tests were two-tailed and carried out at 95% CI. P-value less than 0.05 was considered statistically significant.

## Ethical approval

The Ethical approval was obtained from the Institutional Review Committee, B.P. Koirala Institute of Health Sciences (Reference Number: 057/076/077-IRC, approval date: 23rd September, 2019) before conducting the study. Verbal Consent was obtained from the contractors to assess construction workers. Written informed consent was obtained from eligible construction workers before enrolling them into the study. The construction workers were informed about the purpose and procedures of the study and were informed that participation was voluntary and hence they could withdraw at any time without further obligations. Confidentiality and anonymity of the participants were maintained and assured throughout the study.

## Results

A total of 456 construction workers were assessed, of which 415 were eligible to be included into the study. Out of 415 participants, 13 (3.1%) declined to participate into the study and response rate of the study was 96.9%.

### Socio-demographic and occupational characteristics

A total of 402 construction workers were included in the study: the socio-demographic and occupational characteristics of the participants are shown in the Table 1. Of 402 participants, 83.8% were males and the age of the participants ranged from 18 to 64 years with the mean age of 31.78±9.49 years. Most of the participants were Hindus (77.2%) followed by Buddhists (18.2). About three-fourth (76.9%) of the participants were married and 16% were illiterate. About one-fifth of the construction workers of the present study were living with less than $1.90 per day (Table 1).

Most of the participants were manual labor representing 41.3% followed by bricklayer accounting for 25.9%. The working hours per week of the participants ranged from 15 to 90 hours per week with an average of 58.35 ± 19.44 hours per week. The resting hours ranged from 0.5 to 3 hours per day with a mean of 1.29 ± 0.44 hours. The experience of the workers ranged from 1 to 40 years with a median of 5 years (Table 1).

**Table 1. Socio-demographic and occupational characteristics of the construction workers (N = 402).**

| Characteristics | n (%) or mean±SD or median(IQR) |
|---|---|
| **Gender** | |
| Male | 337 (83.8) |
| Female | 65 (16.2) |
| **Age (in years)** | 31.78±9.49 |
| **Nationality** | |
| Nepali | 399 (99.3) |
| Indian | 3 (0.7) |
| **Ethnicity** | |
| Chhetri | 65 (16.2) |
| Brahmin | 55 (13.7) |
| Newar | 51 (12.7 |
| Tamang, Sherpa, and Bhote | 70 (17.4) |
| Magar | 50 (12.4) |
| Rai and Limbu | 19 (4.7) |
| Kami/Damai/Badi/Gaine | 20 (5.0) |
| Madhesi Brahmin and Janajati | 7 (1.7) |
| Tharu | 49 (12.2) |
| Other (Gurung, Jirel, thami, chepang, Majhi etc) | 16 (4.0) |
| **Religion** | |
| Hindu | 311 (77.4) |
| Buddhist | 73 (18.2) |
| Islam | 2 (0.5) |
| Kirat | 12 (3.0) |
| Christian | 4 (1.0) |
| **Education** | |
| No education | 64 (15.9) |
| Some primary | 104 (25.9) |
| Completed primary | 45 (11.2) |
| Some secondary | 74 (18.4) |
| Completed secondary | 66 (16.4) |
| Above secondary | 49 (12.2) |
| **Marital Status** | |
| Married | 309 (76.9) |
| Unmarried | 87(21.6) |
| Widowed | 4 (1.0) |
| Separated | 2 (0.5) |
| **Poverty** | |
| Below poverty line | 80 (19.9) |
| Above poverty line | 322 (80.1) |
| **Types of construction work** | |
| Manual labor | 166 (41.3) |
| Bricklayers | 104 (25.9) |
| Internal finish worker | 59 (14.7) |
| Armature fixing | 49 (12.2) |
| Painter | 14 (3.5) |
| Electrician | 10 (2.5) |
| **Work per week (hours),** mean±SD | 58.35 ± 19.44 |
| **Rest per day (hours),** mean±SD | 1.29 ± 0.44 |

(*Continued*)

**Table 1.** (Continued)

| Characteristics | n (%) or mean±SD or median(IQR) |
|---|---|
| **Age at joining construction industry (year),** mean±SD | 23.29±7.10 |
| Less than 18 year | 127 (31.6) |
| More than 18 year | 275 (68.4) |
| **Work experience in the construction sector (years),** median(IQR) | 5 (3, 11) |
| One year | 51 (12.7) |
| 2–10 years | 246 (61.2) |
| 11–20 years | 81 (20.1) |
| Above 20 years | 24 (6.0) |

N: Total frequency; n: frequency; %: Percentage; SD: Standard deviation.

IQR: Interquartile Range.

## Prevalence of low back pain

Table 2 shows the overall distribution of LBP. One-year prevalence of LBP among the construction workers was 52.0% (95% CI: 47.0–57.0) accounting highest prevalence among female, 72.3% (95%CI: 59.8–82.7) in comparison to males, 48.1% (95% CI: 42.6–53.6). The one-month, one-week and point prevalence were all higher among females in comparison to males.

## Factors associated with LBP

Univariate logistic regression analysis revealed significant factors associated with LBP were gender, age, marital status, poverty, perceived enough sleep, sleep quality, presence of comorbidity, work experience, anxiety and job insecurity (Table 3).

Multivariate logistic regression in Table 4 revealed that the odds of having LBP was higher among females (aOR = 2.42; 95%CI: 1.12–5.23), those living below poverty-line (aOR = 2.35; 95%CI: 1.32–4.19), those with more than five years of work experience (aOR = 1.66; 95%CI: 1.01–2.73) and those with intermediate sleep quality (aOR = 2.06; CI: 1.03–4.11).

## Protective measures used by construction workers

The majority of the participants without LBP (95.0%) in the last one year did not use any protective measures to prevent LBP. Only 5.0% of participants used some protective measures of which 4.0% used "patuka" (special piece of cloth worn around the waist) and remaining 1.0% did exercise.

The majority of the participants with LBP (61.2%) did nothing against LBP in past one year. Remaining 28.7% of participants with LBP used "patuka", 8.6% took medication prescribed by doctors, 7.2% took medication from the pharmacy, 2.4% had physiotherapy and 1.4% did the exercise to manage LBP.

**Table 2.** Prevalence of low back pain among construction workers (N = 402).

| Prevalence of low back pain | Prevalence n[% (95% CI)] | | |
|---|---|---|---|
| | **Overall (N = 402)** | **Male (N = 337)** | **Female (N = 65)** |
| One-year | 209 [52.0 (47.0–57.0)] | 162 [48.1 (42.6–53.6)] | 47 [72.3 (59.8–82.7)] |
| One-month | 130 [32.3 (27.8–37.2)] | 98 [29.1 (24.3–34.2)] | 32 [49.2 (36.6–61.9)] |
| One-week | 115 [28.6 (24.2–33.3)] | 85 [25.2 (20.7–30.2)] | 30 [46.2 (33.7–59.0)] |
| Point prevalence | 92 [22.9 (18.9–27.3)] | 65 [19.3 (15.2–23.9)] | 27 [41.5 (29.4–54.4)] |

**Table 3. Univariate logistic regression to determine association of socio-demographic, lifestyle, occupational and psychosocial factors with LBP (N = 402).**

| Variables | Low back pain, n (%) | | Crude Odd's ratio (95% CI) | p-value |
|---|---|---|---|---|
| | **Present** | **Absent** | | |
| **Socio-demographic factors** | | | | |
| **Sex** | | | | |
| Male (ref) | 162 (48.1) | 175 (51.9) | 1 | |
| Female | 47 (72.3) | 18 (27.7) | **2.82 (1.57–5.06)** | <0.001 |
| **Age** | | | | |
| 18–30 years (ref) | 90 (42.5) | 122 (57.5) | 1 | |
| 30–40 years | 75 (60.5) | 49 (39.5) | **2.08 (1.32–3.26)** | 0.002 |
| Above 40 years | 44 (66.7) | 22 (33.3) | **2.71 (1.52–4.84)** | 0.001 |
| **Marital status** | | | | |
| Unmarried (ref) | 24 (27.6) | 63 (72.4) | 1 | |
| Married | 182 (58.9) | 127 (41.1) | **3.76 (2.23–6.34)** | <0.001 |
| **Religion** | | | | |
| Hindu (ref) | 169 (54.3) | 142 (45.7) | 1 | |
| Buddhism | 34 (46.6) | 39 (53.4) | 0.72 (0.44–1.22) | 0.233 |
| Other (Islam, Kirat, Christian) | 6 (33.3) | 12 (66.7) | 0.42 (0.15–1.15) | 0.091 |
| **Poverty** | | | | |
| Below poverty line | 53 (66.3) | 27 (33.8) | **2.09 (1.25–3.49)** | 0.004 |
| Above poverty line (ref) | 156 (48.4) | 166 (51.6) | 1 | |
| **Lifestyle factors and comorbidities** | | | | |
| **Comorbidities** | | | | |
| Absent(ref) | 167 (48.3) | 179 (51.7) | 1 | |
| Present | 42 (75.0) | 14 (25.0) | **3.22 (1.70–6.10)** | <0.001 |
| **Obesity** | | | | |
| Normal (ref) | 100 (50.0) | 100 (50.0) | 1 | |
| Underweight | 15 (42.9) | 20 (57.1) | 0.75 (0.36–1.55) | 0.436 |
| Overweight | 32 (50.8) | 31 (49.2) | 1.03 (0.59–1.82) | 1.000 |
| Obese | 62 (59.6) | 42 (40.4) | 1.48 (0.91–2.38) | 0.097 |
| **Current smoking** | | | | |
| No (ref) | 112 (51.6) | 105 (48.4) | 1 | |
| Yes | 97 (52.4) | 88 (47.6) | 1.03 (0.70–1.53) | 0.870 |
| **Current alcohol** | | | | |
| No (ref) | 99 (49.5) | 101 (50.5) | 1 | |
| Yes | 110 (54.5) | 92 (45.5) | 1.22 (0.82–1.81) | 0.320 |
| **Sleep duration** | | | | |
| 3–4 hours | 3 (60.0) | 2 (40.0) | 5.19 (0.80–33.80) | 0.432 |
| 5–6 hours | 26 (70.3) | 11 (29.7) | **3.29 (1.34–8.09)** | 0.131 |
| 7–8 hours | 113 (47.7) | 124 (52.3) | 1.13 (0.57–2.22) | 0.222 |
| 9 or more hours (ref) | 67 (63.9) | 56 (45.5) | 1 | |
| **Perceived enough sleep** | | | | |
| Yes (ref) | 161 (46.3) | 187 (53.7) | 1 | |
| No | 48 (88.9) | 6 (11.1) | **9.29 (3.88–22.28)** | <0001 |
| **Sleep quality** | | | | |
| Good (ref) | 135 (43.8) | 173 (56.2) | 1 | |
| Intermediate | 39 (69.6) | 17 (30.4) | **2.94 (1.59–5.42)** | 0.001 |
| Poor | 35 (92.1) | 3 (7.9) | **14.9 (4.50–49.66)** | <0.001 |
| **Occupational factors** | | | | |
| **Type of construction work** | | | | |
| Manual labor (ref) | 96 (57.8) | 70 (42.2) | 1 | |

(*Continued*)

**Table 3.** (Continued)

| Variables | Low back pain, n (%) | | Crude Odd's ratio (95% CI) | p-value |
|---|---|---|---|---|
| | **Present** | **Absent** | | |
| Bricklayer | 56 (53.8) | 48 (46.2) | 0.85 (0.52–1.39) | 0.521 |
| Internal finish worker | 29 (49.2) | 30 (50.8) | 0.71 (0.39–1.28) | 0.250 |
| Armature fixing | 25 (51.0) | 24 (49.0) | 0.76 (0.40–1.44) | 0.399 |
| Painter and electrician) | 3 (12.5) | 21 (87.5) | **0.10 (0.03–0.36)** | <0.001 |
| **Working hours per day** | | | | |
| 8 hours or less (ref) | 88 (56.1) | 69 (43.9) | 1 | |
| 9 to 10 hours | 50 (45.5) | 60 (54.5) | 0.65 (0.40–1.07) | 0.089 |
| 11 and above | 71 (52.6) | 64 (47.4) | 0.87 (0.55–1.38) | 0.554 |
| **Work experience** | | | | |
| 5 years or less (ref) | 84 (41.6) | 118 (58.4) | 1 | |
| More than 5 years | 125 (62.5) | 75 (37.5) | **2.34 (1.57–3.49)** | <0.001 |
| **Psychosocial factors** | | | | |
| **Depression** | | | | |
| Absent (ref) | 170 (51.1) | 163 (48.9) | 1 | |
| Present | 39 (56.5) | 30 (43.5) | 1.25 (0.74–2.10) | 0.408 |
| **Anxiety** | | | | |
| Absent (ref) | 161 (49.5) | 164 (50.5) | 1 | |
| Present | 48 (62.3) | 29 (37.7) | **1.69 (1.01–2.81)** | 0.043 |
| **Stress** | | | | |
| Absent (ref) | 172 (51.6) | 164 (48.8) | 1 | |
| Present | 37 (56.1) | 29 (43.9) | 1.22 (0.72–2.07) | 0.469 |
| **Satisfaction** | | | | |
| High (ref) | 153 (49.7) | 155 (50.3) | 1 | |
| Low | 56 (59.6) | 38 (40.4) | 1.49 (0.93–2.39) | 0.093 |
| **Work-family life balance** | | | | |
| High | 133 (50.0) | 133 (50.0) | 1 | |
| Low | 76 (55.9) | 60 (44.1) | 1.27 (0.84–1.92) | 0.264 |
| **Job insecurity** | | | | |
| Low (ref) | 104 (47.3) | 116 (52.7) | 1 | |
| High | 105 (57.7) | 77 (42.3) | **1.52 (1.02–2.26)** | 0.037 |
| **Hostile work environment** | | | | |
| No (ref) | 178 (50.7) | 173 (49.3) | 1 | |
| Yes | 31 (60.8) | 20 (39.2) | 1.51 (0.83–2.74) | 0.23 |

N: Total frequency; n: frequency; %: percentage; CI: Confidence interval; ref: reference group.

Bold crude odds ratio indicates significant at 5% level of significance.

## Discussion

This study aimed to find out the prevalence of LBP, factors associated with LBP among construction workers and to find out measures taken to prevent and manage LBP.

### Prevalence of low back pain

The overall one-year prevalence of LBP among construction workers was 52.0%. Our findings are consistent with the previous cross-sectional questionnaire based studies that reported similar one year prevalence of LBP among construction workers [12,13]. In contrast, Reddy *et al*

**Table 4. Multivariate logistic regression analysis to determine independent predictor of LBP (N = 402).**

| Variables | Category | Adjusted Odds Ratio (95% CI) | p-value |
|---|---|---|---|
| **Gender** | Male (ref) | 1 | |
| | Female | **2.42 (1.12–5.23)** | **0.024** |
| **Age** | 18–30 (ref) | 1 | |
| | 30–40 | 1.32 (0.78–2.27) | 0.299 |
| | Above 40 | 1.52 (0.76–3.06) | 0.241 |
| **Poverty** | Above poverty line(ref) | 1 | |
| | Below poverty line | **2.35 (1.32–4.19)** | **0.004** |
| **Comorbidity** | Absence(ref) | 1 | |
| | Presence | **2.18 (1.02–4.65)** | **0.045** |
| **Perceived enough sleep** | Yes(ref) | 1 | |
| | No | **4.93 (1.56–15.60)** | **0.007** |
| **Sleep quality** | Good(ref) | 1 | |
| | Intermediate | **2.06 (1.03–4.11)** | **0.041** |
| | Poor | 3.89 (0.91–16.69) | 0.067 |
| **Type of construction worker** | Manual labor(ref) | 1 | |
| | Bricklayer | 1.02 (0.54–1.93) | 0.955 |
| | Internal finish worker | 0.78 (0.37–1.65) | 0.518 |
| | Armature fixing worker | 1.63 (0.76–3.47) | 0.208 |
| | Painter/electrician | **0.188 (0.05–0.72)** | **0.014** |
| **Work experience** | 5 years or less(ref) | 1 | |
| | More than 5 years | **1.66 (1.01–2.73)** | **0.045** |
| **Anxiety** | Absence(ref) | 1 | |
| | Presence | 1.33 (0.73–2.42) | 0.355 |
| **Job insecurity** | Low(ref) | 1 | |
| | High | 0.84 (0.49–1.44) | 0.522 |
| **Job satisfaction** | High(ref) | 1 | |
| | Low | 0.83 (0.44–1.59) | 0.578 |
| **Constant** | - | 0.401 | 0.007 |

CI: Confidence interval; ref: reference group.

Bold adjusted odds ratio indicates significant at 5% level of significance.

[31] and Kaneda *et al* [32] reported lower prevalence of LBP with 20.8% and 29.3% respectively among the construction workers. Bodhare *et al* [33] *and* Araújo *et al* [34] reported the highest one-year prevalence of LBP with 92.0% and 71.4% respectively among construction workers compared to present study.

One-year prevalence of LBP among male construction workers in the current study was 48.1%. Our finding are similar with the cross-sectional studies conducted by Alghadir *et al* [11] and Ueno *et al* [8] among male construction workers that reported 50.0% and 53.2% respectively. Telaprolu *et al* [13] reported one-year prevalence of LBP to be 44.1% among women construction workers which was much lower compared the present study.

The prevalence of LBP in the general population of low-income countries ranged from 0 to 16% [35] which was far lower than the findings of the present study (52.0%) among the construction workers. The study conducted in Nepal found 36.2% one-year prevalence of LBP among farmers [36] which was less than the result of the present study. Though both agriculture and construction works involved hard physical labor and mechanical nature, the LBP is more common among construction workers than farmers.

## Risk factors for low back pain

**Socio-demographic risk factors.** One-year prevalence of LBP was higher among females compared to male construction workers (LBP: 72.0% vs. 48.0%). The findings showed an agreement with several studies conducted among construction workers and the general population [8,12,32,33,37–39]. In contrast, there was no difference in the prevalence of LBP between males and females in a study carried among Japanese construction workers [32]. The difference in the prevalence of LBP may be due to sex differences that could be related to gonadal steroid hormones such as testosterone and estradiol modulate sensitivity to analgesia and pain [40].

The age of the participants with LBP was significantly higher than those without LBP. Increasing age was significant for the outcome of LBP in the current study. The role of age on LBP was also reported by Holmström *et al* [12], Kaneda *et al* [32], and Ueno *et al* [8]. This is because, as population ages, LBP increases substantially due to the deterioration of the intervertebral discs in older people [21].

In the present study, the married have 3.76 times the odds of LBP compared to unmarrried. This finding was congruent with the finding made by Kaneda *et al* where unmarried participants were 0.70 times less likely to have LBP [32]. This relationship might be because married participants have a significantly higher age than unmarried participants.

In the current study, LBP was found to be significantly higher among construction workers living below poverty line. This association can be explained in two possible ways. One is an inability to perform productive work due to the presence of LBP may be driving workers into poverty. Another is poor nutrition due to poverty might be leading to LBP.

**Lifestyle risk factors.** *Alcohol intake.* Several studies have not listed alcohol intake as a risk factor of LBP but a study had shown that continued alcohol intake was related to the deterioration of muscle strength and appearance of histological injury to muscle [41]. Hence, alcohol consumption was taken as one of the associated factors for LBP, however, the current study did not find any significant association of LBP with alcohol consumption and is similar to the study by Ueno *et al* [8].

*Smoking.* Researchers don't take smoking as a cause of musculoskeletal pain but as a confounding factor [8] as smoking is associated with job dissatisfaction, job insecurity, anxiety, stress, and depression [42]. A population-based study showed that smokers were 1.23 times likely to develop LBP than non-smokers [38]. In contrast, the present study did not find any association between smoking and LBP among construction workers. Further detailed studies are necessary to determine the causal relationship between LBP and smoking.

*Obesity.* Study conducted by Shiri R *et al* showed increased overweight is associated with lumbosacral reticular pain [43] and an increase in BMI is associated with lumbar disc herniation which is the important cause of LBP [44]. Deyo *et al* showed a significant increase in the prevalence of LBP with an increase in body mass index [45]. Our study, similar to study conducted by Chung *et al* [46], failed to find any association between LBP and obesity among construction workers.

*Sleep.* Pain has been reported to have bi-directional relationship with sleep—pain hinders sleep and sleep disturbances reduce the pain threshold and mental capacity to manage pain [47]. This study failed to determine any association with LBP.

**Occupational risk factors.** In the present study, the prevalence of LBP was significantly lower among painter and electrician in comparison to manual labors and other types of construction workers. This might be due to the difference in the nature of work and working posture among the construction workers. Manual labors, bricklayers and armature fixing workers are more prone to working postures like bending heavily with one's trunk, bending and

twisting simultaneously with one's trunk, a bent and twisted posture for long periods, and making repetitive movements with the trunk or exposed to vibrations which are known occupational risk factors for LBP [21].

Work experience was significantly higher among workers with LBP compared to those without LBP in this study which is similar to the findings of Kaneda et al [32].

There have been various reports on the strong relationship between the duration of work and the prevalence of LBP [32,48,49]. In contrast to these studies, our study did not find any association with the duration of work.

**Psychosocial risk factors.** *Depression, anxiety, and stress.* Psychosocial factors have been found to play an important role in the development of LBP [50]. Previous studies stated that psychological distress was associated with LBP among various groups of working population [50–52] including construction workers [53]. Depression and anxiety are considered as an internalizing type of psychological distress [54]. Some researchers suggested an association of psychological distress and LBP might be due to the influence of work-related psychosocial factors [50–53]. In the present study, there was a significant association and LBP with anxiety. In agreement with our findings, a study by Frymoyer *et al and* Abolfazl *et al* also found an association between depression and anxiety with LBP [55,56].

*Job Insecurity.* Some studies have indicated musculoskeletal disorder including LBP as the damaging effect of job insecurity [30]. It is believed that mental strain linked with job insecurity may indirectly lead to "physiological vulnerability" which, in turn, may contribute to LBP [27]. Hence job insecurity was taken as independent variable in the present study. We found significant association between job insecurity and LBP among construction workers in univariate analysis but there existed no significant association after adjusting other variables.

*Work-family life imbalance.* Work-family imbalance is postulated to cause mental strain which in turn results in muscle tension or other physiological processes that might exacerbate LBP [57]. In addition, work-family life imbalance is believed to drain psychological and physical resources leading to unhealthy behaviours, including alcohol and tobacco use and decreased leisure-time physical activity which is expected to cause LBP [58]. But this study could not find significant association of work-family balance with LBP among construction workers.

*Hostile work environment and job satisfaction.* Researcher have hypothesized that job dissatisfaction and exposure to hostile work environment leads to mental strain which in turn alters biochemical and physiological processes of pain perception leading to musculoskeletal pain including LBP [30,57]. But our study could not find independent association of exposure to hostile work environment and job dissatisfaction with LBP among construction workers.

## Protective measures

A study through observation of intra-abdominal pressure and the lumbosacral compression force confirmed that on wearing "patuka" or lumbar supporter might be accountable for the low incidence of LBP [59]. Very few construction workers, 28.7% among those with LBP and 4.0% among those without LBP were using "patuka" to prevent or manage LBP. Similar to the present study, construction workers of Japan also poorly use lumbar supporter as protective equipment for LBP [32]. According to a study by Shrestha *et al*, safety practices of Nepalese construction projects, mainly the use of personal protective equipment, is gradually growing [60]. Though the use of personal protective equipment (PPE) is growing in the context of Nepal, construction workers in the present study were poorly using PPE such as "patuka" or belt to prevent LBP.

## Limitations

Even though this study tried best to find the prevalence of LBP and factors associated with LBP among the construction workers, it is not free from limitations. Because of the cross-sectional nature of the study, the directionality of the risk associations cannot be established. Several factors like awkward, static and dynamic working posture; times of bending, pushing, pulling, dragging, carrying and holding; rare factors like osteoporosis, prolonged corticosteroid use, vertebral infections, and tumours were not collected. The assessment of the psychosocial risk factors like job insecurity, job satisfaction, work-family life balance and exposure to a hostile work environment was done in this study using single items for each psychosocial domain.

## Conclusion

It can be concluded that the prevalence of LBP in the past year was high among construction workers. Factors like gender, poverty, co-morbidity, sleep quality, and work experience were found to be independently associated with the presence of LBP in the past year. A high proportion of construction workers did nothing to prevent or manage LBP. The findings of the present study are applicable to develop public health and occupational health strategies, programs and guidelines in Nepal to counteract the problem of LBP among construction workers and save the potential of labor workforce.

## Acknowledgments

We would like to acknowledge all the people who directly or indirectly contributed to the present study and all the study participants who shared their valuable time with us.

## Author Contributions

**Conceptualization:** Bikram Adhikari, Anup Ghimire, Sangita Majhi.

**Data curation:** Bikram Adhikari.

**Formal analysis:** Bikram Adhikari, Niroj Bhandari.

**Funding acquisition:** Bikram Adhikari, Anup Ghimire.

**Investigation:** Bikram Adhikari, Anup Ghimire, Nilambar Jha, Archana Shrestha, Aarju Niraula.

**Methodology:** Bikram Adhikari, Anup Ghimire, Archana Shrestha, Roshan Dhakal, Sangita Majhi, Antesh Kumar Pandit, Niroj Bhandari.

**Project administration:** Bikram Adhikari, Antesh Kumar Pandit.

**Resources:** Bikram Adhikari, Nilambar Jha, Rajendra Karkee, Archana Shrestha, Roshan Dhakal, Aarju Niraula, Sangita Majhi, Antesh Kumar Pandit.

**Software:** Bikram Adhikari, Archana Shrestha, Antesh Kumar Pandit.

**Supervision:** Bikram Adhikari, Anup Ghimire, Nilambar Jha, Rajendra Karkee, Archana Shrestha, Roshan Dhakal.

**Validation:** Bikram Adhikari, Anup Ghimire, Nilambar Jha, Rajendra Karkee, Archana Shrestha, Roshan Dhakal, Aarju Niraula.

**Visualization:** Bikram Adhikari, Anup Ghimire, Roshan Dhakal, Aarju Niraula, Sangita Majhi, Niroj Bhandari.

**Writing – original draft:** Bikram Adhikari, Nilambar Jha, Rajendra Karkee, Archana Shrestha, Sangita Majhi, Niroj Bhandari.

**Writing – review & editing:** Bikram Adhikari, Anup Ghimire, Rajendra Karkee, Archana Shrestha, Roshan Dhakal, Aarju Niraula, Niroj Bhandari.

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
