## [Decision Letter · Decision Letter 0]

10 Mar 2021

PONE-D-21-04117

Factors associated with low back pain among construction workers in Nepal: A cross-sectional study

PLOS ONE

Dear Dr. Adhikari,

Thank you for submitting your manuscript to PLOS ONE. After careful consideration, we feel that it has merit but does not fully meet PLOS ONE’s publication criteria as it currently stands. Therefore, we invite you to submit a revised version of the manuscript that addresses the points raised during the review process.

We look forward to receiving your revised manuscript.

Kind regards,

Pranil Man Singh Pradhan

Academic Editor

PLOS ONE

Journal Requirements:

2.We suggest you thoroughly copyedit your manuscript for language usage, spelling, and grammar. If you do not know anyone who can help you do this, you may wish to consider employing a professional scientific editing service.  

3. In your Methods section, please provide a justification for the sample size used in your study, including any relevant power calculations (if applicable).

Reviewers' comments:

Reviewer's Responses to Questions

**Comments to the Author**

1. Is the manuscript technically sound, and do the data support the conclusions?

Reviewer #1: Yes

Reviewer #2: Partly

2. Has the statistical analysis been performed appropriately and rigorously? 

Reviewer #1: Yes

Reviewer #2: Yes

3. Have the authors made all data underlying the findings in their manuscript fully available?

Reviewer #1: Yes

Reviewer #2: Yes

4. Is the manuscript presented in an intelligible fashion and written in standard English?

Reviewer #1: Yes

Reviewer #2: No

5. Review Comments to the Author

Reviewer #1: This manuscript addresses the factors associated with low back pain among construction workers. The manuscript is well written and structured. The title and abstract are appropriate for the content of the paper. All the data are well understood. Furthermore, the manuscript is well constructed but the introduction needs to be revised but the analysis is well performed.

Reviewer #2: a. In abstract: CLBP – short form has been used abruptly without prior mention

b. Reference pattern in 49-56 lines are (?). It is abruptly starting from 21? And it starts directly from sentence line 64?

c. Line 103- why DASS was used? Nothing mentioned in objectives about it.

d. Line 113- why mentioned abruptly about CLBP. Nowhere has been ever talked about these types. Never mentioned in introduction clearly except abruptly mentioned once.

e. Line 140- define others (you have those answers for others… mentioned here )

f. Line 198- what do you mean by age limitation? As you have had exclusion criteria for age.

g. Results: Line 197- why did you assess 456 when they did not meet the criteria? Explain sample size calculation

h. Please explain the enrollment process of the study participants clearly.

i. Who are others in Ethnicity? Correct it

What do you mean? (J-m)

j. Family Size 5 (4, 6)

k. Dependent family members 3 (2, 4)

l. Monthly income (participant) N.Rs.25000 (20000, 25250)

m. Monthly family income N.Rs. 45000 (30000, 55000)

n. Age at joining construction industry (year): it carries great value so it would be best if you could show table of age joining for below 18 years because 127 is a good number.

o. Table 4: amazing that age is not the predictor of LBP .Maybe you should try regrouping the age. Generally age is the predictor.

p. It was difficult to understand difference between LBP and CLBP in your study. Please elaborate.

q. Line 401- Health seeking behavior for low back pain was very poor among construction workers. As nowhere in your study you have mentioned the pain score so you cannot make this statement. Pain scale would have been better to be used and explore grades of LBP. I think this study need this to make your study more valid.

r. You have so many factors included in tables. Please make it smart including only needy variables. Many variables are unnecessarily shown. You have too many unnecessary variables which are not needed for this study and have no direct connections. So for this article those variables are not needed.

s.

t. Chronic back pain has good connection with Depression. Here mostly you have used LBP not CLBP. So you should use this cautiously when needed.

u. No variables which show real relations with LBP such as “working posture”, times of bending, pushing, pulling, dragging, carrying and holding are not in this study. So as topic suggests “the factors associated” is not supported in real.

v. LBP is also strongly associated with the bed quality where workers sleep. No variables related to it show a bias too.

6. PLOS authors have the option to publish the peer review history of their article (what does this mean?). If published, this will include your full peer review and any attached files.

Reviewer #1: **Yes: **Dr.Nikita Bhattarai

Reviewer #2: No

---

## [Author Response · Author response to Decision Letter 0]

8 Apr 2021

Thank you reviewer 1 and reviewer 2 for your fruitful comments. 

The answers to the reviewer’s (Reviewer 2) comments are addressed individually below:

• Reference pattern in 49-56 lines are (?). It is abruptly starting from 21? And it starts directly from sentence line 64?

The references are updated as per the guideline of the journal

• Line 103- why DASS was used? Nothing mentioned in objectives about it.

DASS is now mentioned in the objectives too

• Line 113- why mentioned abruptly about CLBP. Nowhere has been ever talked about these types. Never mentioned in introduction clearly except abruptly mentioned once.

CLBP is a Chronic Low back pain. We’ve assessed CLBP among construction workers but with discussion with co-authors we decided to remove CLBP from the present manuscript as it is confusing. We will submit supplementary article of this manuscript if accepted.

• Line 140- define others (you have those answers for others… mentioned here )

Other may include Glaziar, dry wall installer, and stone wall builder. Other is removed from the sentence as only construction workers mentioned in the list are enrolled in the present study. 

• Line 198- what do you mean by age limitation? As you have had exclusion criteria for age

Yes, <18 years age is the exclusion criteria. For the purpose of documentation, we included the percentage of ineligibility along with the reasons.

We’ve assessed total 456 workers of which 5.7% were of age less than 18 years working in construction worker and 3.3% of the participants could not be communicated due to language barrier (speaking Bhojpuri or Indian language).

• Results: Line 197- why did you assess 456 when they did not meet the criteria? Explain sample size calculation

I’ve added sample size calculation section. We visited all municipalities and checked for presence of construction sites and workers. And all who meet eligibility criteria were included in the study which resulted to count of 456 participant who were assessed.

• Please explain the enrollment process of the study participants clearly.

It is clarified

• Who are others in Ethnicity? Correct it

Gurung, Jirel, thami, chepang, Majhi etc were other ethnicity. It is now mentioned in the table.

• What do you mean? (J-m) 

Family Size 5 (4, 6)

Dependent family members 3 (2, 4)

Monthly income (participant) N.Rs.25000 (20000, 25250)

Monthly family income N.Rs. 45000 (30000, 55000)

They are median and Interquartile range. It is now clarified in the table

• Age at joining construction industry (year): it carries great value so it would be best if you could show table of age joining for below 18 years because 127 is a good number.

The mean and standard deviation of age of joining construction industry presents in the manuscript.

• Table 4: amazing that age is not the predictor of LBP .Maybe you should try regrouping the age. Generally age is the predictor.

I tried to regroup and analyze previously but could not conclude age as predictor.

• It was difficult to understand difference between LBP and CLBP in your study. Please elaborate.

CLBP is removed from the study with the discussion between co-authors. We planned to submit another supplementary article related to CLBP

• Line 401- Health seeking behavior for low back pain was very poor among construction workers. As nowhere in your study you have mentioned the pain score so you cannot make this statement. Pain scale would have been better to be used and explore grades of LBP. I think this study need this to make your study more valid.

The statement is removed from the manuscript. Our study assessed the pain scale among participants with CLBP so Pain scale related information in not added in the manuscript.

• You have so many factors included in tables. Please make it smart including only needy variables. Many variables are unnecessarily shown. You have too many unnecessary variables which are not needed for this study and have no direct connections. So for this article those variables are not needed.

I’ve removed some less important variables from the tables.

• Chronic back pain has good connection with Depression. Here mostly you have used LBP not CLBP. So you should use this cautiously when needed.

CLBP is removed from the study with the discussion between co-authors. We planned to submit another supplementary article related to CLBP

• No variables which show real relations with LBP such as “working posture”, times of bending, pushing, pulling, dragging, carrying and holding are not in this study. So as topic suggests “the factors associated” is not supported in real.

They are included in the limitation of the study.

• LBP is also strongly associated with the bed quality where workers sleep. No variables related to it show a bias too.

Our study assess the sleep quality and perceived enough sleep in order to address the bed quality which is related to LBP.

---

## [Decision Letter · Decision Letter 1]

19 May 2021

Factors associated with low back pain among construction workers in Nepal: A cross-sectional study

PONE-D-21-04117R1

Dear Dr. Adhikari,

We’re pleased to inform you that your manuscript has been judged scientifically suitable for publication and will be formally accepted for publication once it meets all outstanding technical requirements.

Kind regards,

Pranil Man Singh Pradhan

Academic Editor

PLOS ONE

Additional Editor Comments (optional):

Reviewers' comments:

Reviewer's Responses to Questions

**Comments to the Author**

1. If the authors have adequately addressed your comments raised in a previous round of review and you feel that this manuscript is now acceptable for publication, you may indicate that here to bypass the “Comments to the Author” section, enter your conflict of interest statement in the “Confidential to Editor” section, and submit your "Accept" recommendation.

Reviewer #1: All comments have been addressed

Reviewer #2: All comments have been addressed

2. Is the manuscript technically sound, and do the data support the conclusions?

Reviewer #1: Yes

Reviewer #2: Yes

3. Has the statistical analysis been performed appropriately and rigorously? 

Reviewer #1: Yes

Reviewer #2: Yes

4. Have the authors made all data underlying the findings in their manuscript fully available?

Reviewer #1: Yes

Reviewer #2: Yes

5. Is the manuscript presented in an intelligible fashion and written in standard English?

Reviewer #1: Yes

Reviewer #2: Yes

6. Review Comments to the Author

Reviewer #1: (No Response)

Reviewer #2: i have seen the artcile with abstract and it is final from my side.

Background: Low back pain (LBP) is the commonest cause of disability throughout the

world. This study aimed to determine the prevalence and factors associated with LBP

among the construction workers in Nepal.

Methods: A community-based cross-sectional study was conducted among the

construction workers working in Banepa and Panauti municipalities of Kavre district,

from September 2019 to February 2020. Data was collected purposively by face-toface interview from 402 eligible participants from the both municipalities using semistructured questionnaire. Mobile-based data collection was done using KoboCollect.

Data were exported to and analysed using R-programming software (R-3.6.2).

Univariate and multivariate logistic regressions were performed. All tests were two

tailed and performed at 95% confidence interval (CI).

Result: One-year prevalence of LBP among construction workers were 52.0% (95%CI:

47.0-57.0). The higher odds of LBP was reported among females [adjusted odds ratio

(aOR) =2.42; 95%CI: 1.12-5.23], those living below poverty-line (aOR=2.35; 95%CI:

1.32-4.19), participants with more than five years of work experience (aOR=1.66;

95%CI: 1.01-2.73) and those with intermediate sleep quality (aOR=2.06; CI: 1.03-

4.11). About 80.0% of construction workers with LBP never seek healthcare services

due to: a) time constraints (90.9%), b) financial constraints (18.1%) and c) fear of

losing wages on seeking healthcare services (40.9%). The majority of the participants

(94.8% among those without LBP and 72.3% among those with LBP) did nothing to

prevent or manage LBP.

Conclusion: The prevalence of LBP in the past one year was high among construction

workers where majority of workers never did anything to prevent or manage LBP.

Therefore, the public health professionals should set up the health promotion,

education, and interventions aimed at increasing awareness on preventive techniques

and predisposing factors of LBP.

7. PLOS authors have the option to publish the peer review history of their article (what does this mean?). If published, this will include your full peer review and any attached files.

Reviewer #1: **Yes: **Nikita Bhattarai

Reviewer #2: No

---

## [Editor Report · Acceptance letter]

21 May 2021

PONE-D-21-04117R1 

Factors associated with low back pain among construction workers in Nepal: A cross-sectional study 

Dear Dr. Adhikari:

I'm pleased to inform you that your manuscript has been deemed suitable for publication in PLOS ONE. Congratulations! Your manuscript is now with our production department. 

Kind regards, 

on behalf of

Dr. Pranil Man Singh Pradhan 

Academic Editor

PLOS ONE